# Dealing with the diagnosis of Parkinson's disease and its implications for couple functioning in the early stage: An interpretative phenomenological analysis

Emilie Wawrziczny[1]*, Bérengère Flinois[2], Emilie Constant[1], Elodie Brugallé[1], Céline Sokolowski[1], Charlotte Manceau[1], Guillaume Baille[2], Luc Defebvre[2], Kathy Dujardin[2], Pascal Antoine[1]

1 University of Lille, CNRS, UMR 9193 – SCALab – Sciences Cognitives et Sciences Affectives, Lille, France,
2 Regional and University Hospital Centre of Lille, Lille, France

* wawrziczny@univ-lille.fr

**Data Availability Statement:** Data cannot be shared publicly because of confidentiality issues and lack of authorization. Data are available from

## Abstract

### Background

For couples facing Parkinson's disease, marital relationships are significantly impacted, even at the early stages of the disease. However, very few studies have explicitly explored the functioning of the couple and how both partners deal with Parkinson's disease. The aim of this study was to explore the experiences and strategies of couples facing Parkinson's disease in the early stage using dyadic interpretative phenomenological analysis.

### Methods

Fifteen couples agreed to participate in the study. Semistructured interviews were conducted with each partner separately regarding his or her individual experience with Parkinson's disease, the couple's history, the impact of the diagnosis on the functioning of the couple, and his or her projections for the future.

### Results

Three higher-order themes emerged from the analyses. The first theme, "Being tested by the diagnosis", highlights 4 dyadic configurations according to the individual's and the couple's capacity for adjustment following the diagnosis: "noncongruent", "collapsed", "relieved" and "avoidant". The second theme, "Talking about everything except the disease", underlines that communication about the disease is often avoided both within the couple and with relatives to protect the persons with Parkinson's disease or respect their wishes. The third theme, "Supporting each other", describes the different levels of harmony between the two partners in the management of daily life and symptoms and their relational impacts.

the EST IV Committee for the Protection of Individuals (contact:1, place de l'Hôpital 67091 STRASBOURG Cedex, Tel : 03.88.11.60.03, Fax : 03.88.11.63.48, E-mail : cpp.est4@chru-strasbourg.fr).

**Funding:** PA, Grant 951CV723, the France Parkinson association, the funders had no role in study design, data collection and analysis, decision to publish, or preparation of the manuscript. PA, "ERC-Generator program: Towards a dyadic approach to neurodegenerative diseases" (TADAND), the foundation I-SITE – Univ. Lille Nord Europe, the funders had no role in study design, data collection and analysis, decision to publish, or preparation of the manuscript.

**Competing interests:** The authors have declared that no competing interests exist.

## Conclusion

These results allow us to better understand the experiences of both partners and to highlight the importance of promoting better acceptance of the diagnosis by persons with Parkinson's disease to allow better communication between partners and with relatives. Such support prevents disease-specific distress and facilitates better adjustment in the later stages of the disease.

## Introduction

Parkinson's disease (PD) is the second most prevalent neurodegenerative disease after Alzheimer's disease [1]. According to the World Health Organization, in 2019, there were over 8.5 million people living with PD. PD is a chronic progressive disease clinically characterized by motor symptoms (e.g., bradykinesia, rigidity, rest tremor) and nonmotor symptoms (e.g., sleep disorders, gastrointestinal symptoms, depression, anxiety, urinary disorders, sexual dysfunction) [2–4]. As the disease progresses, persons with PD (PwPDs) become increasingly dependent, have to manage medication and have to deal with loss of control and anxiety about the future [5–8]. Their social life is also impacted due to misunderstanding and stigmatization from the social community [9]. Moreover, they feel shame and frustration due to communication impairments, which may lead to isolation [10].

Therefore, spouses are essential to maintaining a reasonable quality of life for PwPDs [7]. Spouses are heavily involved in managing the disease. They face increased responsibilities and psychosocial challenges (i.e., attending care appointments, monitoring medications, assisting the PwPD with daily activities) [11–13]. They most frequently encourage and motivate the PwPD to stay active and engaged as a daily living coach while considering their own challenges as secondary [7, 14]. Indeed, due to their involvement in the care of the PwPD, they lose valued activities and become socially isolated [12], which impacts their health and well-being [11]. Spouses often feel an increased burden and distress because of caregiving situations [15, 16], and they also suffer greatly from seeing the PwPD suffer and struggle [12, 14].

The disease of one partner of a couple generates significant individual changes and distress for PwPDs and their spouses and creates changes and stress for their relationship. Couples feel overwhelmed by the management of the disease and the activities of daily living [17]. PD is associated with poor marital adjustment. Indeed, couples have to address changes in the role of each partner, disruptions in identity and relationship markers and communication difficulties [12, 18, 19]. PwPDs talk about moving from a romantic partnership to a roommate-type relationship or a caregiver–patient or parent–child-like dyad [20]. Partners share fewer social and leisure activities [14, 20]. Both partners also become less responsive to the other partner's needs [21], and they are confronted with changes in sexuality due to limitations in movement or erectile dysfunction [20, 22]. Finally, PwPDs feel uncertainty and fear about whether the partner will remain in the relationship after diagnosis [20]. In return, the quality of the relationship can also impact spouse burden as well as spouses' and PwPDs' quality of life [5, 23]. Indeed, high mutuality protects spouses against increased role strain and burden, and it is associated with less depression for both partners and lower symptom severity for PwPDs [24, 25]. Some partners have also reported a strengthening of the relationship, greater closeness, more reciprocity [14, 17] and reinforcement or validation of their mutual dedication [20]. Mavandadi et al. [26] also showed that perceived positive impacts of the disease, such as personal growth, were related to better conjugal quality of life for both partners.

While the disease has significant impacts on the marital relationship, very few studies have explicitly explored the functioning of the couple and addressed how both conjugal partners cope with PD [27]. Moreover, previous studies have highlighted the importance of distinguishing stages of disease progression so as not to confuse different situations regarding the duration of the disease or the intensity of symptoms [7, 27] and to have a meaningful understanding of the needs of couples facing PD and provide better assistance [28, 29]. The early period is still underexplored, perhaps because previous studies have suggested that the role transition for both partners (the need for more assistance by PwPDs and the caregiving role for spouses) begins in the moderate stage of the disease [28, 30] and increases in the advanced stage [31]. However, previous studies have identified motor and nonmotor symptoms, self-identity changes, relationship changes, and catastrophic thinking related to death in the early stages of the disease that may be detrimental to patients' functional status and the sense of wellbeing for both partners [6, 29, 32–34]. Moreover, while nonmotor symptoms precede the diagnosis by several years [35, 36], the diagnosis is often experienced as a "bomb" or a shock [37], and its acceptance may be made difficult by misdiagnosis rates of up to 20–30% in the early stages [38]. Finally, for couples confronted with PD, the level of mutuality (i.e., positive relationship quality) declines beginning in the early stage, particularly for younger couples [29, 39].

The objective of this study is to explore the experiences of couples in which one partner is at an early stage of PD, i.e., in the few years after diagnosis when symptoms are still mild and well controlled by treatments, as well as to better understand individual and conjugal implications for both partners.

## Method

### Participants

The inclusion criteria were as follows: heterosexual couples who had been living together for at least 5 years, in which one partner was at an early stage of PD evaluated by the Hoehn and Yahr stage (Hoehn & Yahr stages 1 and 2; [40, 41] and did not suffer from another neurological or psychiatric disease; PwPDs and their spouse who were aged between 40 and 80 years to avoid atypical and advanced forms of PD and to ensure their capacity to physically and cognitively communicate and to coconstruct their experience. These criteria were controlled by the referring neurologist of each patient before the proposition of the study.

The interviews were analyzed using the interpretive phenomenological analysis (IPA) method developed by Smith et al. [42]. For this method, the literature recommends a small number of participants, ranging from 1 to 20, to master the whole corpus of data. We chose a sample of heterosexual dyads to ensure homogeneity of the sample.

Table 1 shows that for 8 of the 15 couples who participated in the study, the PwPD was a man. The mean age of the PwPDs was 58 years (SD = 10.04), and the mean age of the partners was 59 years (SD = 11.19). The average length of time since diagnosis was 1 year (SD = 0.64). The mean duration of the couple relationship was 36 years (SD = 9.62).

### Procedure

The couples were recruited from October 2018 to December 2019 through the Neurological and Movement Disorders Department of the Lille University Hospital (France). Couples who met the inclusion criteria received an information letter accompanied by an oral explanation of the study by the patient's referring psychologist or neurologist. With the partners' agreement, they were called by interviewers, and the first appointment was scheduled to take place at their home or at the hospital. Before the interview began, each partner was required to sign

**Table 1. Characteristics of couples.**

| Couple | Pseudonym | | Age | | Gender | | Number of children | Couple relationship duration (years) | Time since diagnosis (years) | Hoehn & Yahr |
|---|---|---|---|---|---|---|---|---|---|---|
| | PwPD | Partner | PwPD | Partner | PwPD | Partner | | | | |
| A | Aurélie | Adrien | 47 | 41 | F | M | 2 | 29 | 2 | 1 |
| B | Bernard | Béatrice | 51 | 52 | M | F | 3 | 34 | 3 | 1 |
| C | Cyril | Cathy | 59 | 60 | M | F | 1 | 40 | 2 | 1 |
| D | David | Doriane | 67 | 68 | M | F | 2 | 51 | 1 | 1 |
| E | Esteban | Emilie | 61 | 58 | M | F | 1 | 37 | 2 | 1 |
| F | Fabienne | Fabrice | 66 | 70 | F | M | 3 | 43 | 2 | 1 |
| G | Gilles | Géraldine | 77 | 75 | M | F | 2 | 56 | 2 | 2 |
| H | Hélène | Henri | 72 | 75 | F | M | 2 | 43 | 1 | 1 |
| I | Isabelle | Isidor | 57 | 57 | F | M | 3 | 38 | 1 | 1 |
| J | Jules | Juliette | 66 | 68 | M | F | 2 | 39 | 1 | 1 |
| K | Kris | Kate | 48 | 46 | M | F | 2 | 20 | 1 | 1 |
| L | Louise | Léon | 53 | 54 | F | M | 2 | 29 | 1 | 1 |
| M | Matthieu | Maude | 50 | - | M | F | 2 | 29 | 1 | 1 |
| N | Natacha | Nicolas | 49 | 53 | F | M | 3 | 30 | 1 | 1 |
| O | Olivia | Oscar | 44 | 44 | F | M | 3 | 26 | 1 | 1 |

Abbreviation: PwPD, person with Parkinson's disease; F, Female; M, Male; -, not collected.

a consent form. Separate semistructured interviews with each partner were conducted simultaneously. The dyadic approach is an interesting way of highlighting commonalities and differences between what each partner experiences and their disagreements. The topics dealt with the partners' individual experiences with PD, the couple's history, the impact of the diagnosis on the functioning of the couple, and the partners' projections for the future (see Table 2). The PwPDs' interviews lasted an average of 60 minutes (SD = 11.62), and the partners' interviews lasted an average of 66 minutes (SD = 13.45). The interviews were recorded using a voice recorder and transcribed in their entirety. Pseudonyms were used to designate the participants,

**Table 2. Interview grid.**

| |
|---|
| 1. What can you tell me about the disease? How did you live it? |
| • How did it start? |
| • How did you experience the diagnosis? |
| 2. Individual adjustments: |
| • How does the disease affect your life? How do you cope with these changes? How do you feel? |
| • According to you, how does your partner live the disease? |
| 3. Couple history and dyadic functioning before the disease: |
| • Can you tell me about your couple relationship? (Meeting, communication, intimacy, evolution) |
| • In your couple story, have you encountered any difficulties? |
| 4. Dyadic adjustments: |
| • How does the disease affect your relationship? |
| • How do you manage these changes with your partner? |
| • Has the disease created a distance between you and your partner? |
| • Do you perceive positive consequences? |
| 5. How do you consider the future with your partner? |

with the same initial given to both partners within a couple (e.g., couple A: Aurélie for the PwPD and Adrien for the partner).

## Ethical issues

The study was approved by the National Ethics Committee (Committee for the Protection of Persons, CPP Est IV, IDRCB 2017-A0261152). The team of the Lille University Hospital that recruited the participants (KD, BF, LD) had the names of the participants but could not make the link with the anonymised interviews. The researchers who conducted the interviews (EC, EB, CS, CM) had access to information that could identify participants during data collection. The other authors (EW, PA, GB) did not know the names of the participants and only had access to the anonymised interviews.

## Data analysis

The purpose of this study was to report the experience of couples confronted with the PD of one partner in the early stage. IPA permits access to the participants' subjective experiences, the meaning that they attributed to a given phenomenon, and the psychological processes [42, 43]. The interviewers, psychologists experienced in clinical interviewing and IPA, were aware of the impact of their subjectivity in this type of analysis. They therefore adopted a reflexive attitude, putting aside their knowledge and assumptions about the participant's experience [44].

According to our research question, the two partners of the couple are concerned with research with a shared and distinctive experience. For this reason, data were collected following a multiperspective design [45]. The interviews were conducted separately with each partner to allow access to more honest and intimate information without fear of hurting the other partner regarding one's experience of the disease and one's view of the couple dynamics.

Following the recommendations of Nizza, Farr, and Smith [46], each interview was analyzed and interpreted by two researchers. They first analyzed each interview individually. The IPA is based on the coconstruction of meaning by the researcher and the participant. During the interview, the participant makes an interpretation of his or her experience. The researcher then analyses the transcript of this interview line by line, focusing on the vocabulary and expressions used by the participant to describe his or her experience and emotions. These salient elements enabled the researcher to access the meaning of the experience for the participant and his or her psychological mechanisms, which were then schematized by the researcher. This double hermeneutic allows the researcher to delve deeper into the participant's discourse by putting aside his or her presuppositions about the participant's experience. The two researchers then shared and discussed their analyses conjointly to enable reflexivity and to prevent potential bias of their own opinions and feelings. They explored the experience of each partner, and they considered the two interviews of each couple together to understand the convergences and differences between partners. These steps were repeated for all 15 couples. At the end of the 15 analyses, the researchers highlighted the salient individual and dyadic phenomena to synthesize the experience of couples confronted with the PD of one partner in the early stage.

## Results

Three themes were identified: "Being tested by the diagnosis", "Talking about everything except the disease", and "Supporting each other" (see Table 3). The first theme describes four conjugal configurations according to their individual and dyadic adjustment capacities after the diagnosis. The second one explores the difficulties encountered by both partners in talking

**Table 3. Summary of themes.**

| Overarching themes | Being tested by the diagnosis | Talking about everything except the disease | Supporting each other |
|---|---|---|---|
| **Subthemes** | *Noncongruent (couples B, H, I, M, N, and O) | * Not talking is protecting | * A harmonious adjustment |
| | *Collapsed (couples A, C, K and L) | * The unsaid spreads outside the couple | * A more difficult adjustment |
| | *Relieved (couples F and G) | | |
| | *Avoidant (couples D, E and J) | | |

about the disease and its psychological repercussions. Finally, the third theme illustrates the different levels of harmonization in managing daily life and symptoms. These themes are detailed and supported by examples from the interviews.

## Being tested by the diagnosis

At the early stage of the disease, the arrival of the first signs of disease is a source of concern more or less shared by both partners. Individual and dyadic adjustment capacities are asked for after the diagnosis according to four configurations: noncongruent (couples B, H, I, M, N, and O), collapsed (couples A, C, K and L), relieved (couples F and G) and avoidant (couples D, E and J).

The "noncongruent" configuration, characterized by noncongruent experiences between the partners, is well illustrated by couple O. For Olivia, the first signs of disease worry her, and she quickly seeks information to understand what is happening to her; she then communicates this information to the more distant and skeptical Oscar. She feels some frustration but interprets Oscar's reaction as part of a necessary time to increase his awareness of the disease. This proactive approach helps prepare her to receive the diagnosis, whereas Oscar receives the news of the diagnosis abruptly, feeling thrown into an uncertain future.

*My wife, who was certainly more worried about her health than I was, had already looked into the networks, into other things [. . .]. She had come across Parkinson's disease [. . .] So she waited for confirmation, which was almost a relief because she was afraid to know if it wasn't something else. For me, it had the other effect, i.e., a snowball effect. I became afraid [. . .] it's things that we can no longer control.*

*(Oscar, partner)*

*I had a slight suspicion, having looked at all the symptoms and finding that I had 80% of them. But when we found out, it's true that he cried quite a bit.*

*(Olivia, PwPD)*

The spouse who has more resources may then become a support for his or her partner, for example Béatrice: "I pull him up [. . .] like a puppet. I say 'come on, come on'".

For the "collapsed" configuration, illustrated by couple C, after a reaction to the first signs that are more or less shared by both partners, the diagnosis provokes an individual emotional collapse followed by dyadic recovery. At first, the partners withdraw into themselves and then

very quickly find themselves united in minimizing, relativizing and reassessing the situation, relying on the couple as a resource.

> *A sledgehammer blow. . .For a week, I didn't dare talk to him about it; he didn't talk to me about it. From time to time, we'd say a word, but it didn't last any longer. We took it upon ourselves; we talked about it together, and then we said, "Come on. We're both here. We're both going to do what we have to do."*
>
> *(Cathy, partner)*
>
> *For a week I was put on anti-anxiety medication [. . .] I was feeling down, I thought, "This is it; it's over [. . .]" Likewise, she took the brunt of it for a week, and then after that. . . she's not the type to let go. . . we're going to fight.*
>
> *(Cyril, PwPD)*

For the "relieved" couples, illustrated by couple G, the first signs are associated with other illnesses in the family history, causing strong anxiety in one or both partners. The fear of being confronted again with difficult situations is a source of tension in the couple. Nevertheless, the diagnosis of PD, perceived as less serious than those illnesses already encountered in the family, is a source of relief for the partners and reduces these tensions.

> *His mum wasn't diagnosed with Parkinson's; it was Alzheimer's. But I can still hear the sound of my mother-in-law's slippers walking, and it reminded me of that, and it made me tense up every time. . .! [. . .] I was afraid to live like that.*
>
> *(Géraldine, partner)*
>
> *Sometimes she was a bit annoyed by my memory lapses, by the fact that I would start a sentence and not finish it.*
>
> *(Gilles, PwPD)*

In 'avoidant' couples, both partners are more or less aware of the first disease manifestations, but after diagnosis, both partners put in place or reinforce strategies to fight and avoid the disease. At the individual level, the partners may question the diagnosis, and the PwPD actively compensates for the symptoms to make them as invisible as possible. At the dyadic level, these efforts are accompanied by a lack of communication about the disease. This dynamic hinders the partners' awareness and adaptation, as witnessed by couple D, and is a source of tension and misunderstanding.

> *I wasn't even sure if it's PD. He says no [. . .] he never really told me he had Parkinson's.*
>
> *(Doriane, partner)*
>
> *Even now, I think I realize only a little bit. . . or I don't want to realize.*
>
> *(David, PwPD)*

## Talking about everything except the disease

The majority of couples describe themselves as open and communicative about family life, decisions to be made and adaptation to daily life. On the other hand, everything concerning

the disease, from the diagnosis to the symptoms and their psychological repercussions, is the subject of more subtle communication at this early stage between respect for intimacy, protection and unsaid.

**Not talking is protecting.** The difficulties are partly concealed or downplayed, and feelings are similar to a "secret garden", as Fabienne put it. This discretion on the part of each partner about his or her own experiences, as couple A testifies, aims to protect the other partner, not to worry or weaken him or her and not burden him or her with more than he or she is already enduring.

*I don't tell her about it because I don't want to worry her more than she already is.*

*(Adrien, partner)*

*There are things I don't tell him about. . . like my leg. I don't tell him that I realize it's not getting better [. . .] I don't want to, maybe to preserve him, not to worry him [. . .] He doesn't talk about it often; I try to talk to him about it from time to time, but I think he doesn't want to show me how it's affecting him. . .*

*(Aurélie, PwPD)*

This discretion is accompanied by increased attention and receptivity to the emotional manifestations of the partner and by capacity to sense, beyond words and in the absence of a dialog, when the other is not well. For Jules and Fabienne, this intuitive mode of communication, without the need to talk to each other to understand each other, is inherent to marital complicity.

However, when emotions are perceived to be overflowing, Aurélie, Béatrice and Maude use various strategies and insist to get their partner to talk. Béatrice explains, "There is music that makes him crack! So sometimes when I feel it, I'll put the music on, and then as soon as I see him crying, I say 'That's it! Go on, say it'''.

For Hélène, her difficulties and feelings are freely and daily evoked and even take up too much space for Henri, who either tries to find subtle diversions with activities or more abruptly ends the conversation. Hélène explains: «Sometimes he understands, and sometimes he tells me: "you're annoying me, you just have to let it go and that's it. . .". . .I don't like that because I think. . . either I'm exaggerating, or he's exaggerating, but I don't understand who's exaggerating. Then, we are not in dispute, we talk about other things and then that's it". This lack of communication about her feelings does not allow Hélène to find the right adjustment and leaves her in a state of misunderstanding. However, she is more secretive about the suicidal thoughts she may have when she projects herself into a physical state unacceptable for her as the disease progresses, judging Henri too "sensitive" to hear them.

**The unsaid spreads outside the couple.** Some partners, such as those in couples B, G, J and O, open up freely to individuals around them, finding these individuals to be more attentive, benevolent, and sometimes even overly protective. Others, such as couples A, C, D, E, H, I, L, M and N, do not inform or a minima (test results, medical appointments) their relatives of their recent diagnosis, by fear of being disappointed, or of disturbing them, or of giving too much space to the disease or of being perceived as an invalid. The behavior of the relatives and the activities are then adapted according to the disease but without mentioning it. When the disease is hidden, PwPDs exhaust themselves hiding their symptoms or try to choose situations that expose them as little as possible. The spouses of PwPDs, such as Adrien, worry that the social environment is prevented from adapting to the real capacities of the person with disease: "With the disease, they (employers) have to give you a break, but, no. . . as they don't know

[. . .] I'm afraid it will hurt her [. . .] It's her choice, I can't go through it". Moreover, this lack of information can sometimes lead to misunderstandings about certain behaviors of the PwPD (i.e., impulsivity. . .) and tensions between PwPD and the relatives that the spouse tries to mediate.

Whether the PwPDs mention the diagnosis openly, talk about it indirectly by referring to it as something else, such as a "lack of dopamine" as Jules does, or attribute the symptoms to another disease, the spouses, such as Doriane, respect this decision and the time that will be needed to discuss it. As Cathy notes, "It is his decision, and I'm not going to go around talking about it either", even if this can lead to uncomfortable situations and prevent partners from confiding in people outside the couple.

*I don't tell anybody, because I have a sister who has Parkinson's disease, and she told me, "It looks like David is shaking". . . so I said, "Yes, but it's because of Lyme disease." I don't dare say it.*

*(Doriane, partner)*

*If someone sees my tremors, I'm not going to tell them it's Parkinson's, I'm going to say it's Lyme disease or something [. . .] I don't want my wife to talk about it.*

*(David, PwPD)*

## Supporting each other

Most couples describe themselves as being united and cohesive in the face of disease, and some draw parallels with their unity during previous life events, e.g., miscarriage, disease, financial problems or death of a relative. Nevertheless, the couples differ in their level of harmony in managing daily life and symptoms.

**A harmonious adjustment.**   Several modes of harmonization can be identified. Some partners, such as Jules and Juliette or Hélène and Henri, adjust their emotional expressions and behaviors based on the needs of the other. Others, such as the partners of couples C, F, K, I and O, take turns emotionally, which Oscar describes as "communicating vessels". Cyril observes, "We both pull each other up, together [. . .] when things aren't going well on his side, I'm there and then, vice versa". This echoes the functioning of couple F (see below). The partners thus experience their relationship as a resource.

*We are very, very different, but we complement each other well, so we appreciate [. . .] that everyone has their moments of difficulty and dreariness, and we are never synchronized. I don't know why, but when I have my moments of dreariness, she generally takes things in hand, and when she has her moments of dreariness, I take things in hand.*

*(Fabrice, partner)*

*Fortunately, he's here because he supports me a lot. . . When he sees that things aren't going well. . . he also feels it when things aren't going well, or I'm not in the mood. . . so we support each other. . .*

*(Fabienne, PwPD)*

**A more difficult adjustment.**   Some couples (A, B, D, E, G, L, M and N) are confronted with different levels of independence management. PwPDs are determined to maintain their independence and their pace of life, sometimes minimizing and pushing their limits. They are then confronted with their partners, who become increasingly invasive, worried, and attentive after the diagnosis to protect them or to fight the disease.

*I think I was always like that and maybe not anymore, because even sometimes She says, "Leave it. I know how to do it". . . I say, "That's true, but you can ask me." She says, "No, I'm not impotent." [. . .] It's true that she does a lot. I tell her, "You do a lot", and she also says, "Oh, I'm tired [. . .] It's OK, I'll rest when I'm dead." [. . .] It was rare that we took a nap, but now, from time to time, we try, and I even push my wife to take a nap for her to really relax.*

*(Adrien, partner)*

*I want to do everything on my own. He realizes that sometimes I have difficulties, so he wants to help me. I say, "No. . .I don't want to, listen to me, I want to remain autonomous. . ." even if I know that sooner or later, there will come a time when I will perhaps, I will certainly lose this autonomy, but as late as possible. . . Now, I'm less and less settled. I always want to do something because I don't want the disease to take hold. . . but he can see it in my face. . . he says, "No, no, no, you stop" [. . .]*

*It annoys me enormously because I've never been like this. I've always been on the move, not listening to myself, and then I realize that the disease is taking over, so I'm obliged to comply, and then I settle down, and then it gets better. . .*

*(Aurélie, PwPD)*

The spouses may then react by imposing themselves on medical appointments, constantly offering help, insistently proposing activities to fight the progression of the disease, or on the contrary, by setting limits for the PwPD to rest. These interventions may give the PwPDs the impression of capitulating to the disease, being deprived of their free will, and being supervised or infantilized. The spouses may then use strategies, as Géraldine says, to conceal their controls and interventions. The adjustment is even more difficult when the PwPD avoids everything related to the disease and that the spouses supplant the PwPD, as Doriane recounts: "He doesn't want to believe that he is ill[. . .] He refuses, he never takes care of his medication, his disease stuff; I'm always the one who has to take care of it." Irritability and tension may emerge between the two partners who deal with their anxiety due to the arrival of disease in their own way without trying to understand the other. These tensions put the couple in difficulty, even though their autonomy is still preserved at this early stage.

## Discussion

The original objective of this study was to specifically explore individual and conjugal implications in the early stage of the disease for couples in which one of the partners had been diagnosed with PD. Indeed, this period is still underexplored when individual and marital disruptions are already present due to the initial symptoms and diagnosis [29, 33, 34, 37]. The analyses of interviews allowed us to observe different individual and conjugal issues concerning the experience of the first signs of disease and diagnosis, communication about the disease and dyadic support.

## The experience of the first signs of disease and the diagnosis

Our results highlighted four configurations according to individual and dyadic experiences and adjustment following the first signs of disease and the diagnosis: "noncongruent", "collapsed", "relieved" and "avoidant". For "relieved" couples, the tension that emerges with the first signs of disease is relieved with the diagnosis. Previous studies have already pointed out that the diagnosis may be a source of relief by naming the condition causing the symptoms despite anxiety about an uncertain future due to the degenerative character of PD [37]. In the case of "avoidant" couples, diagnosis quickly leads to avoidance and struggle, consequently restricting communication about the disease. This lack of communication leads to tension and misunderstanding between both partners. Manne et al. [47] showed that mutual avoidance of discussing disease-related issues is associated with higher distress for both partners and with conjugal dissatisfaction. For "collapsed" and "noncongruent" couples, anxiety following the diagnosis is linked to a feeling of loss of control, powerlessness and uncertainty. The support that partners give to each other enables them to find the means to adjust individually and as a couple.

## Communication about the disease

The results showed that the majority of couples describe communication about daily life to be very smooth but communication about their experience of the disease to be much more difficult. Indeed, disease-related communication in couples is not always equivalent to general intracouple communication [48].

The majority of participants in this study prefer to keep their feelings to themselves and try not to show what they feel out of pride or to protect their partner. They then externalize when the partner forces a dialog, either when they are alone or when they are outside the conjugal sphere. However, it has been shown that clear and open communication about the disease is important for both partners' well-being and permits successful management of its biopsychosocial effects, while avoidance and holding back thoughts and emotions are associated with poor outcomes for both partners, including lower relationship satisfaction and greater distress [17, 47–51]. Perceptions of communication about the disease should be assessed by clinicians with the Couples' Illness Communication Scale (CICS) soon after diagnosis. This assessment would allow the referral of early communication interventions, such as the Early Diagnosis Dyadic Intervention (EDDI) program developed by Whitlatch et al. [52] adapted to PD, to prevent disease-specific distress in the long term [48].

The results also highlighted that in an effort not to worry or annoy others and not to appear fragile or less efficient, some PwPDs decide not to talk about their disease to their relatives, which leads them to control their symptoms in public or to adapt to situations. Indeed, PwPDs fear losing their jobs, being stigmatized, being embarrassed, being treated as ill, and being perceived as incapable, which can threaten their sense of identity [17, 53]. They thus attempt to show others and themselves that they are still capable of performing tasks, try to hide shaking from others, often without success and avoid social interactions. Fostering better acceptance of the disease would allow PwPDs to reduce their struggle against the disease and help them be able to better express their needs in terms of assistance from their spouses so that they can better adjust.

Moreover, spouses understand and respect the decision of PwPDs not to talk about the disease. However, this prevents the family or professional environment from adapting to the disease (demands, activities, etc.). Spouses are also placed in the uncomfortable position of having to lie or not being able to confide in family and friends about their experiences. Family mediation sessions may be encouraged to open discussion with family, help other family

members gain a better understanding of the disease, and assess available support [54]. Indeed, when PwPDs talk about the disease openly with their family and friends, they often receive real emotional support and more attention and kindness.

## Dyadic support

In the face of this adversity, the different ways of giving and receiving help can either bring the two partners closer together or create tension. Martin [53] and Vikström et al. [55] already showed that in the face of disengagement from activities by PwPDs, the closeness of caregivers can sometimes be an asset to the relationship and a catalyst for the closeness of both partners. Coping with the challenges of the disease allows partners to affirm their mutual commitment to each other and consolidate their relationship [17]. This study also showed that adversities encountered in the relationship history have bonded and strengthened the couple. The disease is then perceived as an additional event that the partners must go through together as a couple. Previous studies on cancer have shown that couples who conceptualize a diagnosis as a "couple stressor" have better marital adjustment than those who see it as an "individual stressor" [56, 57]. 'We' awareness may be considered a protective factor for conjugal adjustment to the disease. A three-step process designed by Skerrett [57] may be proposed to couples with an 'I' awareness to build and nourish a "we" awareness. In the first step, couples may be supported to build a "we" awareness; in the second step, couples may build awareness of the disease history of each partner; and in the third step, couples may serve to nourish and reinforce the "we" as a resource.

Nevertheless, support is not always easy for PwPDs to accept. PwPDs may find that their spouses become more attentive and anxious with the diagnosis, whereas they want to preserve their independence, sometimes to the point of not listening to themselves and pushing their own limits. This struggle with symptoms can cause greater fatigue and, in turn, an accentuation of the symptoms, which worries spouses. Indeed, spouses are aware of the identity issues of PwPDs but worry about health consequences, and they feel powerless [53]. This worry may lead spouses to make remarks, supervise, monitor or try to set limits for PwPDs, which can be a source of tension. Support is then perceived as a hindrance to PwPDs' involvement in daily mutual commitments [55], sometimes giving PwPDs the impression of being infantilized or capitulating to the disease. Therefore, partners may be faced with the dilemma of supporting their partners without threatening their sense of independence and contributing to a feeling of being controlled or infantilized. This dilemma is especially constraining since it is not always easy to find a compromise between helping and not helping [11] and to know when PwPDs want to act independently and when they want help [53] since the perception of symptoms, abilities and help is sometimes different between the two partners [49]. It would be interesting to support spouses in better adapting their helping behaviors to the PwPD's symptoms, using more encouragement, modeling, and reinforcement, and supporting PwPDs in their own choices. Moreover, communication about the disease would be increased, and discussions about how to cope with daily activities would be encouraged to make joint decisions. Such support would promote a sense that both partners are working together to cope with the disease [58].

## Limitations

Several limitations of our study may limit the generalization of the results. First, the sample was limited to heterosexual couples. Second, to participate in this study, both partners were voluntary and had to agree and make the decision together to participate. This implies that either the couples who agreed were not the couples with a high level of distress and the greatest

difficulty in communicating, they were then less inhibited and shared their experience with greater ease; or at the contrary, they experienced greater distress and difficulties than most couples in this situation, and they used the interview as a way of expressing them.

## Conclusion

This study is the first to focus on the early stage of the disease experience of spouses and PwPDs using an IPA dyadic approach. The interviews allowed us to better understand the experiences of both partners and to highlight the importance of better support as soon as the first symptoms appear of disease or just after the diagnosis to facilitate better adjustment in the later stages of the disease. Indeed, early adjustment in the disease experience is the strongest predictor of the later adjustment [59]. It is also important to promote better acceptance of the diagnosis by PwPDs to reduce their struggle against the disease and to allow better communication with their partners and with their family. Improved communication would allow better adjustment among relatives concerning the help to be given. Communication about the disease between the two partners can be evaluated early to refer early communication interventions to prevent disease-specific distress in the long term. Spouses may also be supported in developing more encouragement and stimulation behaviors to the PwPD's symptoms. Couples may be supported in building, nourishing and reinforcing a "we" awareness, which is considered a resource and a protective factor. Family mediation sessions may be encouraged to gain them a better understanding of the disease and organize support according to the support available. In future qualitative research, the best approach might be to combine individual interviews with dyadic interviews conducted with both spouses together to assess their interactions [49]. It would also be interesting to explore the experiences of homosexual couples and to examine, in the context of a longitudinal study, what happens when symptom severity increases.

## Acknowledgments

We thank all couples for their participation in this study and the Lille Centre of Excellence for Neurodegenerative Disorders (LICEND) for their support.

## Author Contributions

**Conceptualization:** Emilie Wawrziczny, Guillaume Baille, Luc Defebvre, Kathy Dujardin, Pascal Antoine.

**Formal analysis:** Emilie Wawrziczny, Bérengère Flinois.

**Funding acquisition:** Pascal Antoine.

**Investigation:** Bérengère Flinois, Emilie Constant, Elodie Brugallé, Céline Sokolowski, Charlotte Manceau.

**Methodology:** Emilie Wawrziczny, Kathy Dujardin, Pascal Antoine.

**Project administration:** Pascal Antoine.

**Resources:** Luc Defebvre, Kathy Dujardin.

**Supervision:** Pascal Antoine.

**Visualization:** Emilie Wawrziczny.

**Writing – original draft:** Emilie Wawrziczny.

**Writing – review & editing:** Emilie Wawrziczny, Bérengère Flinois, Emilie Constant, Elodie Brugallé, Céline Sokolowski, Charlotte Manceau, Guillaume Baille, Luc Defebvre, Kathy Dujardin, Pascal Antoine.

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
