## [Decision Letter · Decision Letter 0]

26 Sep 2023

PONE-D-23-17328Dealing with the diagnosis of Parkinson's disease and its implications for couple functioning in the early stage: an interpretative phenomenological analysisPLOS ONE

Dear Dr. wawrziczny,

Thank you for submitting your manuscript to PLOS ONE. After careful consideration, we feel that it has merit but does not fully meet PLOS ONE’s publication criteria as it currently stands. Therefore, we invite you to submit a revised version of the manuscript that addresses the points raised during the review process. Please submit your revised manuscript by Nov 09 2023 11:59PM. If you will need more time than this to complete your revisions, please reply to this message or contact the journal office at plosone@plos.org. Please include the following items when submitting your revised manuscript:A rebuttal letter that responds to each point raised by the academic editor and reviewer(s). You should upload this letter as a separate file labeled 'Response to Reviewers'.A marked-up copy of your manuscript that highlights changes made to the original version. You should upload this as a separate file labeled 'Revised Manuscript with Track Changes'.An unmarked version of your revised paper without tracked changes. You should upload this as a separate file labeled 'Manuscript'.If applicable, we recommend that you deposit your laboratory protocols in protocols.io to enhance the reproducibility of your results. Protocols.io assigns your protocol its own identifier (DOI) so that it can be cited independently in the future. For instructions see: https://journals.plos.org/plosone/s/submission-guidelines#loc-laboratory-protocols. Additionally, PLOS ONE offers an option for publishing peer-reviewed Lab Protocol articles, which describe protocols hosted on protocols.io. Read more information on sharing protocols at https://plos.org/protocols?utm_medium=editorial-email&utm_source=authorletters&utm_campaign=protocols.

We look forward to receiving your revised manuscript.

Kind regards,

Steve Zimmerman, PhD

Associate Editor, PLOS ONE

Journal Requirements:

**Additional Editor Comments:** The manuscript has been evaluated by three reviewers, and their comments are available below.All the reviewers are very positive about your manuscript, but have several requests that mostly relate to clarification and additional details.

Could you please revise the manuscript to carefully address the concerns raised?

Reviewers' comments:

Reviewer's Responses to Questions

**Comments to the Author**

1. Is the manuscript technically sound, and do the data support the conclusions?

Reviewer #1: Yes

Reviewer #2: Yes

Reviewer #3: Yes

2. Has the statistical analysis been performed appropriately and rigorously? 

Reviewer #1: N/A

Reviewer #2: N/A

Reviewer #3: N/A

3. Have the authors made all data underlying the findings in their manuscript fully available?

Reviewer #1: Yes

Reviewer #2: Yes

Reviewer #3: Yes

4. Is the manuscript presented in an intelligible fashion and written in standard English?

Reviewer #1: Yes

Reviewer #2: Yes

Reviewer #3: Yes

5. Review Comments to the Author

Reviewer #1: The authors examine and important, understudied topic that is highly relevant to people with PD, clinicians managing PD, and contributes to our broader understanding of couple functioning following a diagnosis of a chronic disease. The paper is well-written and clearly describes the phenomenon of interest. I have some minor suggestions to improve the clarify of this paper.

Methods:

- Why did you limit the couples to heterosexual couples only?

- Review some wording (e.g. page 7, line 117) there is an extra word

- Can you provide the interview guides? How did you develop them? Did you test the guides at all before conducting the interviews?

- Did you have the researchers conducting the analysis do any kind of reliability testing?

- Can you provide a reflexivity statement?

Results:

- The presentation of themes and quotes is very effective. I appreciate that you provide both partner and PwPD perspectives next to each other. To maintain consistent organization, it might be helpful to have the care partner quotes on the right side or left side for each text box.

Discussion:

- The discussion provides a great overview of the study and how it fits in with the existing literature. I think your suggestion to screen for communication early on in the disease is very important. I am wondering if you had looked into whether or not couples in this sample were undergoing couples therapy or support group participation?

- The limitations you describe are important to note. I think it it also important to comment on bias from your sampling approach?

Table 1:

- Does the "-" mean not collected or refused to answer?

Reviewer #2: The manuscript describes a dyadic interpretative phenomenological study with couples living with Parkinson’s disease. It is an excellently written paper, with clear objectives and findings which offer new insights into the delivery of care and services for couples, rather than an isolated provision by individual.

The abstract provides a clear summary of the study and its findings.

Introduction

A good summary of the problems couples living with Parkinson’s disease face is provided. It builds a good rationale for exploring the wellbeing of couples and the impact of PD on couples, rather than simply on the individual living with it.

There is a clear research objective stated.

Method

Recruitment procedures are clearly described and appropriate permissions were obtained.

There is quite a large sample for an IPA study.

The table of themes generated in the analysis should be presented at the beginning of the results section.

Results

I really like the presentation of the both partners’ words alongside each other. Although interviews were conducted separately, this is a really nice way of bringing their accounts back together.

The interpretative commentary alongside the data provides valuable insight into the couples’ experiences and offers a rich interpretation, the kind anticipated in an IPA study.

Discussion

There is a good discussion of findings against the existing literature. This draws on existing interventions and services available, with recommendations for improving them.

The limitations are carefully considered, alongside the benefits of the way the study was conducted, so we get a nice, balanced view.

Reviewer #3: Thank you for the qualitative study that supplements the missing information oriented to dealing with the diagnosis of Parkinson's disease and its implications for couple functioning.

In order to strengthen the quality of the qualitative study, I would like to ask the authors for:

- clarification of information about Sampling, access and recruitment of participants;

- revising the table number in line 101 ("Table 2 shows that the mean age..."), it shows Table 1;

- specific naming of the steps of the analysis of the life experience of the participants according to the IPA in the Data analysis section;

- addition of data about ensuring the rigor and validity of the qualitative study;

- more precise description of the limitations of the study without providing a description of the dyadic approach (which I recommend to include in the description of the methodology of the qualitative study) and without suggestions for further research (which I recommend to include in the conclusion of the qualitative study);

- clarification of the conclusions of the conducted qualitative study for health care aimed at helping patients with Parkinson's disease and their partners deal with Parkinson's disease and qualitative research in this field.

6. PLOS authors have the option to publish the peer review history of their article (what does this mean?). If published, this will include your full peer review and any attached files.

Reviewer #1: No

Reviewer #2: No

Reviewer #3: No

---

## [Author Response · Author response to Decision Letter 0]

19 Oct 2023

Reviewer #1: The authors examine and important, understudied topic that is highly relevant to people with PD, clinicians managing PD, and contributes to our broader understanding of couple functioning following a diagnosis of a chronic disease. The paper is well-written and clearly describes the phenomenon of interest. I have some minor suggestions to improve the clarify of this paper.

Methods:

- Why did you limit the couples to heterosexual couples only?

We chose a sample of heterosexual dyads to ensure the homogeneity of the sample. We added these sentences in the Method section and in the Limitations. We added the prospect of studying the experiences of homosexual couples within Conclusion (as suggested by another reviewer).

In the Method section :

We chose a sample of heterosexual dyads to ensure homogeneity of the sample.

In the Limitations section :

First, the sample was limited to heterosexual couples.

In the Conclusion :

It would also be interesting to explore the experiences of homosexual couples

- Review some wording (e.g. page 7, line 117) there is an extra word

The extra word was removed :

The topics dealt with the partners’ individual experiences with PD, the couple’s history, the impact of the diagnosis on the functioning of the couple, and the partners’ projections for the future.

- Can you provide the interview guides? How did you develop them? Did you test the guides at all before conducting the interviews?

We added the interview grid in Table 2. We did not test this grid before conducting the interviews but we developed it on the basis of our many years of experience in qualitative interviewing and IPA analysis, as shown in the references below:

Antoine, P., Flinois, B., Doba, K., Nandrino, J. L., Dodin, V., & Hendrickx, M. (2018). Living as a couple with anorexia nervosa: A dyadic interpretative phenomenological analysis. Journal of Health Psychology, 23(14), 1842-1852.

Constant, E., Brugallé, E., Wawrziczny, E., Sokolowski, C., Manceau, C., Flinois, B., ... & Antoine, P. (2022). Relationship dynamics of couples facing advanced-stage Parkinson’s disease: a dyadic interpretative phenomenological analysis. Frontiers in Psychology, 12, 770334.

Manceau, C., Constant, E., Brugallé, E., Wawrziczny, E., Sokolowski, C., Flinois, B., ... & Antoine, P. (2023). Couples facing the “honeymoon period” of Parkinson's disease: A qualitative study of dyadic functioning. British Journal of Health Psychology, 28(2), 366-382.

Wawrziczny, E., Antoine, P., Ducharme, F., Kergoat, M. J., & Pasquier, F. (2016). Couples' experiences with early-onset dementia: An interpretative phenomenological analysis of dyadic dynamics. Dementia, 15(5), 1082-1099.

Wawrziczny, E., Pasquier, F., Ducharme, F., Kergoat, M. J., & Antoine, P. (2016). From ‘needing to know’to ‘needing not to know more’: an interpretative phenomenological analysis of couples' experiences with early‐onset Alzheimer's disease. Scandinavian journal of caring sciences, 30(4), 695-703.

Wawrziczny, E., Corrairie, A., & Antoine, P. (2021). Relapsing-remitting multiple sclerosis: an interpretative phenomenological analysis of dyadic dynamics. Disability and rehabilitation, 43(1), 76-84.

- Did you have the researchers conducting the analysis do any kind of reliability testing?

There is no index of reliability in the IPA method, which is based on the theory of interpretation and involves the co-construction of meaning by the researcher and the participant. Nevertheless, the two researchers shared and discussed their analyses conjointly to enable reflexivity and to prevent potential bias of their own opinions and feelings.

We added this paragraph for more clarity:

The IPA is based on the coconstruction of meaning by the researcher and the participant. During the interview, the participant makes an interpretation of his or her experience. The researcher then analyses the transcript of this interview line by line, focusing on the vocabulary and expressions used by the participant to describe his or her experience and emotions. These salient elements enabled the researcher to access the meaning of the experience for the participant and his or her psychological mechanisms, which were then schematized by the researcher. This double hermeneutic allows the researcher to delve deeper into the participant's discourse by putting aside his or her presuppositions about the participant's experience. The two researchers then shared and discussed their analyses conjointly to enable reflexivity and to prevent potential bias of their own opinions and feelings. 

- Can you provide a reflexivity statement?

We added a little more about the reflexivity point: 

 The interviewers, psychologists experienced in clinical interviewing and IPA, were aware of the impact of their subjectivity in this type of analysis. They therefore adopted a reflexive attitude, putting aside their knowledge and assumptions about the participant's experience (44).

This double hermeneutic allows the researcher to delve deeper into the participant's discourse by putting aside his or her presuppositions about the participant's experience. The two researchers then shared and discussed their analyses conjointly to enable reflexivity and to prevent potential bias of their own opinions and feelings.

Results:

- The presentation of themes and quotes is very effective. I appreciate that you provide both partner and PwPD perspectives next to each other. To maintain consistent organization, it might be helpful to have the care partner quotes on the right side or left side for each text box.

For greater consistency, we have placed partner quotes on the left side for each text box.

Discussion:

- The discussion provides a great overview of the study and how it fits in with the existing literature. I think your suggestion to screen for communication early on in the disease is very important. I am wondering if you had looked into whether or not couples in this sample were undergoing couples therapy or support group participation?

We don't have this information, and this was not raised spontaneously during the interviews, which may be explained by the short time since the diagnosis.

- The limitations you describe are important to note. I think it it also important to comment on bias from your sampling approach?

We modified the Limitations section considering our sampling approach:

Several limitations of our study may limit the generalization of the results. First, the sample was limited to heterosexual couples. Second, to participate in this study, both partners were voluntary and had to agree and make the decision together to participate. This implies that either the couples who agreed were not the couples with a high level of distress and the greatest difficulty in communicating, they were then less inhibited and shared their experience with greater ease; or at the contrary, they experienced greater distress and difficulties than most couples in this situation, and they used the interview as a way of expressing them.

Table 1:

- Does the "-" mean not collected or refused to answer?

We added under the table 1 :

«-, not collected»

 

Reviewer #2: The manuscript describes a dyadic interpretative phenomenological study with couples living with Parkinson’s disease. It is an excellently written paper, with clear objectives and findings which offer new insights into the delivery of care and services for couples, rather than an isolated provision by individual.

The abstract provides a clear summary of the study and its findings.

Introduction

A good summary of the problems couples living with Parkinson’s disease face is provided. It builds a good rationale for exploring the wellbeing of couples and the impact of PD on couples, rather than simply on the individual living with it. There is a clear research objective stated.

Method

Recruitment procedures are clearly described and appropriate permissions were obtained.

There is quite a large sample for an IPA study.

The table of themes generated in the analysis should be presented at the beginning of the results section.

We moved the Table of themes at the beginning of the results section and we added a paragraph to introduce it :

Three themes were identified: “Being tested by the diagnosis”, “Talking about everything except the disease”, and “Supporting each other” (see Table 3). The first theme describes four conjugal configurations according to their individual and dyadic adjustment capacities after the diagnosis. The second one explores the difficulties encountered by both partners in talking about the disease and its psychological repercussions. Finally, the third theme illustrates the different levels of harmonization in managing daily life and symptoms. These themes are detailed and supported by examples from the interviews.

Results

I really like the presentation of the both partners’ words alongside each other. Although interviews were conducted separately, this is a really nice way of bringing their accounts back together.

The interpretative commentary alongside the data provides valuable insight into the couples’ experiences and offers a rich interpretation, the kind anticipated in an IPA study.

Discussion

There is a good discussion of findings against the existing literature. This draws on existing interventions and services available, with recommendations for improving them.

The limitations are carefully considered, alongside the benefits of the way the study was conducted, so we get a nice, balanced view.

 

Reviewer #3: Thank you for the qualitative study that supplements the missing information oriented to dealing with the diagnosis of Parkinson's disease and its implications for couple functioning.

In order to strengthen the quality of the qualitative study, I would like to ask the authors for:

- clarification of information about Sampling, access and recruitment of participants;

To clarify information about sampling, we completed the paragraph concerning Participants:

The inclusion criteria were as follows: heterosexual couples who had been living together for at least 5 years, in which one partner was at an early stage of PD evaluated by the Hoehn and Yahr stage (Hoehn & Yahr stages 1 and 2; 41,42) and did not suffer from another neurological or psychiatric disease; PwPDs and their spouse who were aged between 40 and 80 years to avoid atypical and advanced forms of PD and to ensure their capacity to physically and cognitively communicate and to coconstruct their experience. These criteria were controlled by the referring neurologist of each patient before the proposition of the study.

- revising the table number in line 101 ("Table 2 shows that the mean age..."), it shows Table 1;

We deleted this reference to avoid repetition with the previous sentence.

- specific naming of the steps of the analysis of the life experience of the participants according to the IPA in the Data analysis section;

We completed the Data analysis section with paragraph for more clarity about the steps of the analysis:

The IPA is based on the coconstruction of meaning by the researcher and the participant. During the interview, the participant makes an interpretation of his or her experience. The researcher then analyses the transcript of this interview line by line, focusing on the vocabulary and expressions used by the participant to describe his or her experience and emotions. These salient elements enabled the researcher to access the meaning of the experience for the participant and his or her psychological mechanisms, which were then schematized by the researcher. This double hermeneutic allows the researcher to delve deeper into the participant's discourse by putting aside his or her presuppositions about the participant's experience. The two researchers then shared and discussed their analyses conjointly to enable reflexivity and to prevent potential bias of their own opinions and feelings. 

- addition of data about ensuring the rigor and validity of the qualitative study;

We added the interview grid in Table 2 which was developed on the basis of our many years of experience in qualitative interviewing and IPA analysis, as shown in the references below:

Antoine, P., Flinois, B., Doba, K., Nandrino, J. L., Dodin, V., & Hendrickx, M. (2018). Living as a couple with anorexia nervosa: A dyadic interpretative phenomenological analysis. Journal of Health Psychology, 23(14), 1842-1852.

Constant, E., Brugallé, E., Wawrziczny, E., Sokolowski, C., Manceau, C., Flinois, B., ... & Antoine, P. (2022). Relationship dynamics of couples facing advanced-stage Parkinson’s disease: a dyadic interpretative phenomenological analysis. Frontiers in Psychology, 12, 770334.

Manceau, C., Constant, E., Brugallé, E., Wawrziczny, E., Sokolowski, C., Flinois, B., ... & Antoine, P. (2023). Couples facing the “honeymoon period” of Parkinson's disease: A qualitative study of dyadic functioning. British Journal of Health Psychology, 28(2), 366-382.

Wawrziczny, E., Antoine, P., Ducharme, F., Kergoat, M. J., & Pasquier, F. (2016). Couples' experiences with early-onset dementia: An interpretative phenomenological analysis of dyadic dynamics. Dementia, 15(5), 1082-1099.

Wawrziczny, E., Pasquier, F., Ducharme, F., Kergoat, M. J., & Antoine, P. (2016). From ‘needing to know’to ‘needing not to know more’: an interpretative phenomenological analysis of couples' experiences with early‐onset Alzheimer's disease. Scandinavian journal of caring sciences, 30(4), 695-703.

Wawrziczny, E., Corrairie, A., & Antoine, P. (2021). Relapsing-remitting multiple sclerosis: an interpretative phenomenological analysis of dyadic dynamics. Disability and rehabilitation, 43(1), 76-84.

We added these two paragraphs concerning the rigor and validity of the qualitative analysis:

The interviewers, psychologists experienced in clinical interviewing and IPA, were aware of the impact of their subjectivity in this type of analysis. They therefore adopted a reflexive attitude, putting aside their knowledge and assumptions about the participant's experience (44).

The IPA is based on the coconstruction of meaning by the researcher and the participant. During the interview, the participant makes an interpretation of his or her experience. The researcher then analyses the transcript of this interview line by line, focusing on the vocabulary and expressions used by the participant to describe his or her experience and emotions. These salient elements enabled the researcher to access the meaning of the experience for the participant and his or her psychological mechanisms, which were then schematized by the researcher. This double hermeneutic allows the researcher to delve deeper into the participant's discourse by putting aside his or her presuppositions about the participant's experience. The two researchers then shared and discussed their analyses conjointly to enable reflexivity and to prevent potential bias of their own opinions and feelings.

- more precise description of the limitations of the study without providing a description of the dyadic approach (which I recommend to include in the description of the methodology of the qualitative study) and without suggestions for further research (which I recommend to include in the conclusion of the qualitative study);

As suggested, the description of the dyadic approach was moved in the Method section :

Separate semistructured interviews with each partner were conducted simultaneously, which allowed us to access more honest and intimate information without fear of hurting and offending the other partner. The dyadic approach is an interesting way of highlighting commonalities and differences between what each partner experiences and their disagreements. 

And we made some modifications in the Limitations section :

Several limitations of our study may limit the generalization of the results. First, the sample was limited to heterosexual couples. Second, to participate in this study, both partners were voluntary and had to agree and make the decision together to participate. This implies that either the couples who agreed were not the couples with a high level of distress and the greatest difficulty in communicating, they were then less inhibited and shared their experience with greater ease; or at the contrary, they experienced greater distress and difficulties than most couples in this situation, and they used the interview as a way of expressing them. 

And suggestions for further research were moved in the Conclusion:

In future qualitative research, the best approach might be to combine individual interviews with dyadic interviews conducted with both spouses together to assess their interactions (49). It would also be interesting to explore the experiences of homosexual couples and to examine, in the context of a longitudinal study, what happens when symptom severity increases.

- clarification of the conclusions of the conducted qualitative study for health care aimed at helping patients with Parkinson's disease and their partners deal with Parkinson's disease and qualitative research in this field.

For more clarification of the conclusions, we added these two paragraphs:

It is also important to promote better acceptance of the diagnosis by PwPDs to reduce their struggle against the disease and to allow better communication with their partners and with their family. Improved communication would allow better adjustment among relatives concerning the help to be given. Communication about the disease between the two partners can be evaluated early to refer early communication interventions to prevent disease-specific distress in the long term. Spouses may also be supported in developing more encouragement and stimulation behaviors to the PwPD’s symptoms. Couples may be supported in building, nourishing and reinforcing a "we" awareness, which is considered a resource and a protective factor. Family mediation sessions may be encouraged to gain them a better understanding of the disease and organize support according to the support available.

In future qualitative research, the best approach might be to combine individual interviews with dyadic interviews conducted with both spouses together to assess their interactions (49). It would also be interesting to explore the experiences of homosexual couples and to examine, in the context of a longitudinal study, what happens when symptom severity increases.

---

## [Editor Report · Decision Letter 1]

30 Oct 2023

Dealing with the diagnosis of Parkinson's disease and its implications for couple functioning in the early stage: An interpretative phenomenological analysis

PONE-D-23-17328R1

Dear Dr. emilie wawrziczny

We’re pleased to inform you that your manuscript has been judged scientifically suitable for publication and will be formally accepted for publication once it meets all outstanding technical requirements.

Kind regards,

Margaret Williams, Ph.D

Academic Editor

PLOS ONE

Additional Editor Comments (optional):

I have reviewed the adjustments made by the authors, which are in line with the requirements detailed by reviewers and I am satisfied that the necessary corrections to this manuscript have been made. I am of the opinion that the interpretative phenomenological analysis design as utilised in this study, is not well known to all researchers, unless they are comfortable with  qualitative research. Having engaged with this article I am satisfied that the authors have more than adequately conducted this IPA study with the requisite rigour, and have meticulously engaged with all concerns from the reviewers, providing a detailed rebuttal letter with all adjustments addressed.  
---

## [Editor Report · Acceptance letter]

6 Nov 2023

PONE-D-23-17328R1 

Dealing with the diagnosis of Parkinson's disease and its implications for couple functioning in the early stage: An interpretative phenomenological analysis 

Dear Dr. Wawrziczny:

I'm pleased to inform you that your manuscript has been deemed suitable for publication in PLOS ONE. Congratulations! Your manuscript is now with our production department. 

Kind regards, 

on behalf of

Professor Margaret Williams 

Academic Editor

PLOS ONE